# Multifactorial Colonization of the Pregnant Woman’s Reproductive Tract: Implications for Early Postnatal Adaptation in Full-Term Newborns

**DOI:** 10.3390/jcm12216852

**Published:** 2023-10-30

**Authors:** Piotr Gibała, Anna Jarosz-Lesz, Zuzanna Sołtysiak-Gibała, Jakub Staniczek, Rafał Stojko

**Affiliations:** 1Chair and Department of Gynecology, Obstetrics and Gynecologic Oncology, Medical University of Silesia, 40-211 Katowice, Polandrsojko@sum.edu.pl (R.S.); 2Neonatology Unit, The Guardian Angels Hospital of the Brothers Hospitallers of St. John of God in Katowice, 40-211 Katowice, Poland

**Keywords:** early postnatal adaptation, newborns, infection in pregnancy, vaginal infection, vaginal microbiome

## Abstract

This retrospective study aimed to investigate the impact of microorganisms identified in the reproductive tract on disorders during the early adaptation period in newborns. A cohort of 823 patients and cervical canal cultures were analyzed to identify the presence of microorganisms. Newborns included in the study were divided into two groups due to the number of pathogens identified in the swab from the cervical canal of the mother. The first group consisted of newborns whose mothers had one pathogen identified (N = 637), while the second group consisted of newborns whose mothers had two or more pathogens identified (N = 186). The analysis of disorders of the early adaptation period included the incidence of respiratory distress syndrome, the number of procedures performed with the use of CPAP, oxygen therapy, antibiotic therapy and parenteral nutrition. Respiratory distress syndrome was more common in group II than in group I (85 vs. 31, *p* = 0.001). In group II, CPAP (63 vs. 21, *p* = 0.001), oxygen therapy (15 vs. 8, *p* = 0.02) and antibiotics were used more frequently (13 vs. 8, *p* = 0.01). The findings of this study revealed that the number of pathogens colonizing the reproductive tract had a significant influence on the early adaptation period in newborns. Multifactorial colonization of the reproductive tract was associated with an increased incidence of infections in newborns and a higher prevalence of acid–base balance disorders. This study highlights the importance of monitoring and addressing the microbial composition of the reproductive tract during pregnancy.

## 1. Introduction

### 1.1. Background

The moment of birth is a rapid transition for the newborn from the intrauterine environment to the external environment. During fetal life, the ambient temperature is constant and optimal, oxygen and nutrients are supplied with the umbilical cord blood and the skin is covered with a protective layer of fetal fluid. Birth, the clamping of the umbilical cord together with thermal and tactile stimulation lead to the aeration of the lungs, the stabilization of regular breathing and gas exchange. An increase in systemic arterial pressure together with a decrease in pulmonary vascular resistance initiates changes in the circulatory system, adapting the heart to work with altered pulmonary circulation. In the following hours of life, the remaining organs of the newborn adapt to the changed conditions of functioning. Postnatal adaptation to extrauterine life is a physiological process and normally does not require medical intervention. However, there are situations that go beyond the limits of physiology, which may manifest themselves throughout the period of early adaptation to the 7th–10th day of a newborn’s life. In Poland, newborns are usually discharged from the hospital after the age of 2 days; therefore, the assessment of a newborn rests with primary care physicians and community midwives, whose role is to notice the moment when the physiology of the neonatal period turns into pathology [1].

Cervico-vaginal infections are not without significance for both the fetus and the newborn. They are still the most common cause of morbidity and mortality in the postnatal period. It has been proven that bacteria commonly colonizing the mother’s genital tract can be transferred to the child’s body both intrauterine and during delivery [2]. The immaturity of the newborn’s immune system increases its susceptibility to intrauterine or perinatal infections [3].

In clinical practice, smears are most often performed to identify group B streptococci (GBS—group B—streptococcal), bacteria that are an important cause of complications in the child in the perinatal period. Based on numerous meta-analyses carried out in the 1990s, the use of perinatal prophylaxis of GBS, which contributes to the reduction of infectious complications in newborns, has been demonstrated [4,5].

The factors contributing to neonatal infections and disorders of the early adaptation period in newborns are: [6,7]

Premature rupture of membranes (especially before 37 weeks of gestation),Drainage of amniotic fluid for more than 18 h,Maternal fever,Chorioamnionitis,Birth of a child with a history of GBS infection,Colonization of GBS during pregnancy, especially urinary tract infection,Maternal diabetes,Prematurity,Low birth weight.

The most common causative agents of early neonatal infections (occurring before 72 h of age) in industrialized countries include group *B streptococci*, *E. Coli*, *Enterobacter*, *S. Aureus* and *S. Pneumoniae* [8]. In developing countries, infections caused by *E.Coli*, *Klebsiellapneumoniae*, *Staphylococcus Aureus* and Acinetobacter predominate [9]. The most common form of early-onset infection is sepsis (EOS—Earlyonsetsepsis), beginning in the first hours of life and often progressing rapidly in newborns < 7 days of age. In addition to sepsis, pneumonia or meningitis may develop, being an isolated infection or complicating sepsis in about 25% of cases [6,7,8,9,10]. Recent studies indicate that bacterial vaginosis is a polymicrobial condition in which the processes occurring between the bacteria colonizing the vagina contribute to the formation of infection [11]. Disturbances of the microbiological balance consisting of a decrease in the physiological flora of *Lactobacilius* spp. and displacing it by pathological microbes, including *Streptococcus agalactie*, *Escherichia coli*, *Haemophilusinfluenze*, *Enterococcusfecalis* and *Peptoniphilus*, contribute to intra-amniotic infections and result in increased infectious complications in the newborn [12].

### 1.2. Objectives

The aim of this study was to analyze the microorganisms identified in the reproductive tract and their impact on disorders of the adaptation period in newborns. The neonatal assessment included the Apgar score, severity of jaundice, respiratory distress, the need for continuous positive airway pressure (CPAP) in the first 48 h of life and laboratory and microbiological tests.

The aim was to check the incidence of adjustment disorders among newborns, depending on the number of microorganisms detected in the reproductive tract of mothers.

## 2. Methods

### 2.1. Study Design and Data Sources

The study received consent from the Bioethics Committee of the Medical University of Silesia in Katowice, Poland, on 8 January 2019, number KNW/022/KB/307/18. Patient records were analyzed retrospectively for this study. The newborn’s condition was based on the results of Apgar scores at 1, 3, 5 and 10 min of life, acid–base balance tests, complete blood count, urine and microbiological tests and inflammatory parameters, i.e., procalcitonin (PCT) and C-reactive protein (CRP) blood levels. We evaluated the need for respiratory support, parenteral nutrition, antibiotic treatment and the occurrence of severe or prolonged jaundice within the first 48 h after birth to assess adaptation issues to extrauterine life.

We analyzed the early adaptation of newborns in both groups to determine the incidence of respiratory disorders. No neonate required intubation or mechanical ventilation; frequency of use CPAP and oxygen therapy procedures were used to compare between groups. We also evaluated the rate of antibiotic therapy, parenteral nutrition and pathological jaundice in both groups.

### 2.2. Settings and Participants

Patients who gave birth between 1 January 2018 and 29 February 2020 were analyzed. The study included a group of 823 pregnant women who gave birth with positive cervical canal culture results from the Department of Gynecology, Obstetrics and Gynecologic Oncology, Medical University of Silesia, Katowice, Poland, and the Neonatology Department of the Hospital of the Order of Bonifraters in Katowice. On the day of admission to the hospital, before the gynecological examination, material for bacteriological examination was collected from the cervical canal in each patient by a gynecologist. Taking a vaginal swab upon admission to the hospital is a standard procedure for every pregnant woman.

The criteria for inclusion in this study were: uncomplicated single pregnancy, culture from the lower genital tract on the day of admission to the ward, natural delivery during the same hospitalization and complete medical documentation. The criteria for exclusion from the study were: cesarean section, multiple pregnancies, complicated pregnancies, pre-eclampsia, eclampsia, gestational diabetes, gestational hypertension, thrombophilia, cholestasis of pregnancy, uterine defects, fetal defects and incomplete medical records. Newborns delivered by cesarean section were excluded from this study to eliminate the possible overlapping of postnatal adaptation issues that may arise during childbirth by this method with disorders resulting from maternal colonization.

Newborns included in the study were divided into two groups due to the number of pathogens identified in the swab from the cervical canal of the mother. The first group consisted of newborns whose mothers had one pathogen identified (N = 637), while the second group consisted of newborns whose mothers had two or more pathogens identified (N = 186). Enrolment is presented in the CONSORT 2010 diagram—Figure 1.

### 2.3. Statistics Methods

The normality of the distributions was checked using the Shapiro–Wilk test. The analysis for normally distributed qualitative unrelated variables was performed using the Chi^2^ test. The analysis for normally distributed unrelated quantitative variables was performed using the student’s *t*-test. Quantitative variables not meeting the criteria of normal distribution were compared using the Mann–Whitney U test. All results were statistically processed in Statistica™ 12 PL software. The significance level of *p* < 0.05 was adopted as the criterion for statistical inference.

## 3. Results

The general characteristics of the study group and the control group are presented in Table 1. Both groups did not differ statistically significantly in terms of all parameters analyzed in Table 1.

The most frequently identified microorganisms in the genital tract in the presented groups were *Candida albicans* and *Streptococcus agalactiae*. All microorganisms identified in the endocervical swabs are listed in Table 2.

Comparing the parameters of the acid–base balance, inflammatory markers and bilirubin, lower pH values were observed in group II than in group I (7.31 ± 0.07 vs. 7.39 ± 0.02). Saturation values were also lower in group II than in group I (78.91 ± 13.1 vs. 83.44 ± 6.91). The concentration of HCO3- was statistically significantly lower in group II than in group I (19.17 ± 3.23 vs. 23.22 ± 2.64). The results of the parameters of inflammation from the first day of life were higher in group II than in group I (8.52 ± 8.02 vs. 7.24 ± 6.12), while statistical significance was observed when comparing the concentration of procalcitonin (1.74 ± 1.11 vs. 0.67 ± 0.52), bilirubin (8.92 ± 4.05 vs. 7.92 ± 3.67) and leukocytes (19.57 ± 7.48 vs. 12.51 ± 4.35) in the complete blood count of newborns. The results are presented in Table 3.

The analysis of disorders of the early adaptation period included the incidence of respiratory distress syndrome and the number of procedures performed with the use of CPAP, oxygen therapy, antibiotic therapy and parenteral nutrition. In addition, the number of newborns diagnosed with pathological jaundice was examined. Respiratory distress syndrome was more common in group II than in group I (85 vs. 31, *p* = 0.001). In group II, CPAP (63 vs. 21, *p* = 0.001), oxygen therapy (15 vs. 8, *p* = 0.02) and antibiotics were used more frequent (13 vs. 8, *p* = 0.01). The results are presented in Table 4.

Infections occurred in 21 neonates and were statistically significantly more common in group II—13 (61.9%)—than in group I: 8 (38.1%), *p* = 0.01. We studied the correlation between premature amniotic fluid leakage in mothers and neonatal infections. However, the results showed no statistical significance (*p* > 0.05). There were no incidents of early sepsis in both groups of neonates. Most common in both groups were infections of undetermined origin: II—8 (61.5%) vs. I—4 (38.5%). This difference was statistically significant (*p* = 0.01). For other types of infections, the differences were not statistically significant.

The relationship between infections was also analyzed statistically using neonates and premature amniotic fluid leakage in the mother; however, the results were not statistically significant (*p* > 0.05).

A division was also made according to the type and number of infections. Infections were more frequently observed in group II than in group I. The most common infections were infections with an undetermined origin—8 (61.5%) vs. 4 (38.5%); this result was statistically significant (*p* = 0.01). No statistically significant differences were found for other types of infection. The results are presented in Table 5.

## 4. Discussion

There are data suggesting the possible intrauterine contact of the fetus with DNA and bacterial metabolites [13]. The results of studies on the importance of providing the child with the microbiota of the reproductive tract during childbirth clearly indicate the beneficial effect of such colonization on postnatal adaptation, the colonization of the newborn’s gastrointestinal tract and the stimulation of its immune system. At the same time, however, colonization with the mother’s bacterial flora may cause infection of the newborn [14].

Among the detected infections, the largest part were infections with an undetermined origin, followed by pneumonia and urinary tract infections. In the presented studies, no case of neonatal sepsis was found. According to the available worldwide data, the incidence of early-onset neonatal sepsis varies between 4 and 22 per 1000 newborns [8]. One of the aims of this study was to identify the relationship between the number of pathogens colonizing the cervical canal and the occurrence of infections in newborns. There were statistically significant differences between group II and group I: 13 (61.9%) vs. 8 (38.1%), *p* = 0.01. When dividing by the type of infection, a higher incidence of infections was also observed in group II. In a meta-analysis, Chan et al. confirmed the correlation between maternal infection and neonatal infection.

In the cited study, the authors analyzed the relationship between a positive smear result and the presence of infection in a newborn and showed a correlation. In the seven studies they reviewed, 5.0% (95% CI 1.9–8.2) of maternal colonization cases had clinical signs of neonatal infection. In this study’s analyzed own material, the percentage of infections in newborns was 2.54%—21 cases of infection were recorded. These results confirm the legitimacy of using smears in women before childbirth, as the pathological bacterial flora of the mother’s reproductive tract has an impact on the occurrence of infection in the newborn [8,15]. The presence of pathological flora in the cervix affects the occurrence of PROM in the mother, which may lead to the development of FIRS in the fetus. This can lead to birth asphyxia or the development of infection in the newborn. The colonization of the genital tract with a diverse pathological flora may also have an immunomodulatory effect on the fetus, as demonstrated by the studies of Gilbert et al. carried out on mice. The coexistence of different types of microorganisms, e.g., *Gardnerella vaginalis*, can stimulate the growth of other strains and cause infections, as evidenced by Gilbert’s research. Antibiotic therapy used during pregnancy may also have an immunomodulatory effect on strains colonizing the reproductive tract. The development of resistance to antibiotics may be associated with an increased rate of infection with drug-resistant strains such as MRSA. In recent years, an increase in the rate of infections in neonatal intensive care units with methicillin-resistant staphylococci has been observed [16,17].

The analysis of disorders of the early adaptation period in a newborn, carried out in our own study, indicates a more frequent occurrence of disorders in group II than in group I. Newborns exposed to many microorganisms present in the mother’s cervical canal more often presented respiratory disorders in the first days of life. Respiratory disorders were present in 85 (10.3%) newborns in group II, while in group I there were 31 (3.7%) cases, *p* = 0.001. These results are confirmed by studies by Edwards et al., who indicate an infectious factor as one of the factors determining respiratory disorders in a newborn [18]. So far, it has been proven that breathing disorders in the early adaptation period affect about 7% of children. A higher percentage of respiratory disorders occurs in the group of newborns born before 37 weeks of gestation and amounts to 34%. Then, it gradually decreases until the 41st week of pregnancy, where it is about 0.5%. However, from year to year, the tendency of respiratory distress syndrome in neonates born at term is observed more and more often [16,17,19,20]. Breathing disorders in newborns from full-term pregnancies occur most often in the form of TTN (Transient Tachypnea of the newborn). This is confirmed by our own research where about 90% of breathing disorders observed in newborns covered by this study were transient breathing disorders. The analysis by Gizzi et al. comparing the use of oxygen therapy and CPAP respiratory support confirms the better effect and greater usefulness of CPAP in the group of newborns with TTN. In the collected research material, respiratory support with CPAP was more often used in group II than in group I, respectively, 63 vs. 21, *p* = 0.001. This is because more TTN cases were reported in the study group.

The acid–base balance parameters checked in the study show that in group I, newborns achieved higher results of saturation, pH and pO_2_ compared to group II. This is also related to the parameters of inflammation examined in children in the first days of life. The levels of procalcitonin and WBC were statistically significantly higher in group II than in group I.

None of the infants observed developed sepsis and only three of the 63 infants requiring respiratory support in group II developed pneumonia. The results of our study suggest the possibility of the prolonged and complicated adaptation to the extrauterine life of full-term infants in the case of the multifactorial colonization of the mother’s reproductive tract. However, in the observed group, prolonged and complicated early adaptation was not associated with the more frequent diagnosis of neonatal infection. The results of our study were similar to those of other centers [18,21]. Considering the widespread early use of antibiotics, the data obtained in our study suggest the vigilant observation of full-term newborns in the event of impaired early adaptation and the use of early-onset sepsis calculators to guide antibiotic management in Term Neonates, rather than the rapid initiation of antibacterial therapy.

This study was a retrospective analysis of the documentation of completed hospitalizations of mother–newborn couples. In the case of suspected infection in the neonate during the first 72 h of life, the scope of examinations depended on the clinical symptoms presented by the child. Not all children underwent the same diagnostic tests, which makes it difficult to compare the two groups of neonatal patients. This study included mother–child pairs in which the bacteriological examination of the genital tract at admission was positive in mothers. The goal was to compare differences in the adaptation of the newborns depending on the number of cultured microorganisms. The limitation of this study is the lack of comparison of the incidence of adaptation disorders and infections in newborns in the culture-positive group to the incidence of the same in newborns of mothers whose bacteriological tests were negative.

In addition, in the group of patients with multifactorial colonization regarding the reproductive tract, an attempt was made to divide into subgroups depending on the configuration of the cultured microorganisms. However, the population in each subgroup was too small to obtain statistical significance; therefore, the results were not credible and the reliable assessment of differences in the quality and severity of neonatal adaptation disorders depending on colonization was impossible.

## 5. Conclusions

The number of pathogens colonizing the reproductive tract affects the early adaptation period in newborns.Polymicrobial colonization of the reproductive tract increases the incidence of acid–base balance disorders in newborns.Multifactorial colonization of the reproductive tract increases the number of infections in newborns. However, it does not affect the incidence of early-onset sepsis or pneumonia in full-term newborns.

## 6. Implications

We understand the importance of conveying the practical implications of our findings. In the revised manuscript, we have expanded our discussion to include possible clinical implications based on our study results. Specifically, we have discussed the potential benefits and challenges of routine screening for cervicovaginal infections and the empirical use of antibiotic prophylaxis for these infections in pregnant women. We believe this will provide clearer guidance to the readers on the relevance of our study to clinical practice.

## Figures and Tables

**Figure 1 jcm-12-06852-f001:**
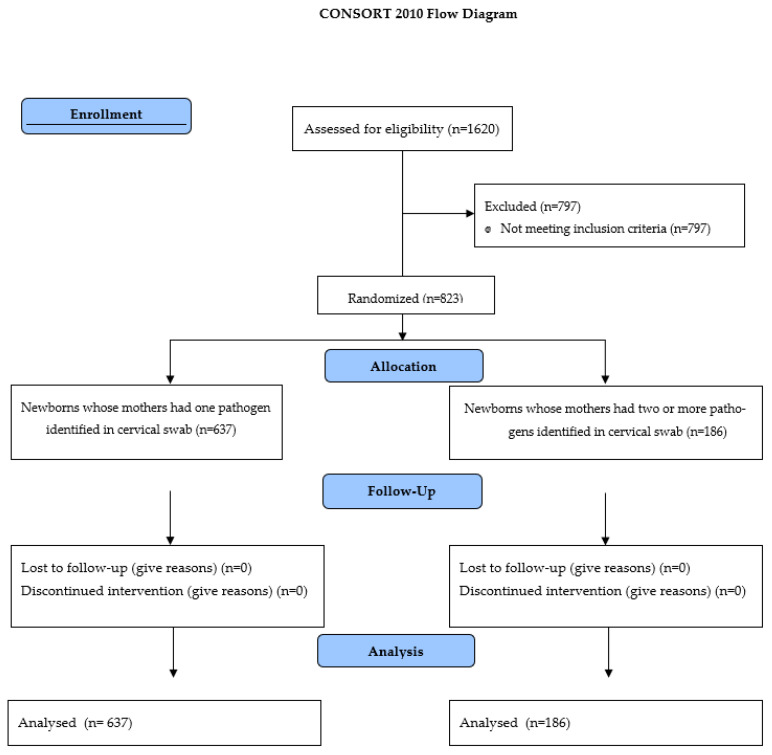
The CONSORT 2010 diagram—Enrolment of participants.

**Table 1 jcm-12-06852-t001:** General characteristics of the study and control groups—analysis with the T-student test of variables: mother’s age, week of pregnancy, number of pregnancies, number of deliveries, BMI, birth weight and Apgar score. *p* value > 0.05—statistically non-significant result.

Variable	Group I (N = 637)	Group II (N = 186)	*p*	Effect Size
Mother’s age	31.03 (SD ± 4.23)	32.1 (SD ± 4.52)	0.54	0.244
Gestational age (weeks)	39.4 (SD ± 1.94)	39.1 (SD ± 1.88)	0.32	0.157
Pregnancy	1.75 (SD ± 0.91)	1.83 (SD ± 0.89)	0.08	0.089
Parity	1.59 (SD ± 0.80)	1.45 (SD ± 0.68)	0.77	0.189
BMI (kg/m^2^)	27.1 (SD ± 3.45)	26.7 (SD ± 2.88)	0.63	0.126
Birth weight (g)	3366 (SD ± 493)	3402 (SD ± 456)	0.37	0.076
Apgar score (1 min)	8.83 (SD ± 0.57)	8.33 (SD ± 1.03)	0.20	0.601
Apgar score (3 min)	9.78 (SD ± 0.77)	9.79 (SD ± 0.92)	0.83	0.012
Apgar score (5 min)	9.37 (SD ± 0.54)	9.23 (SD ± 0.72)	0.44	0.220
Apgar score (10 min)	9.63 (SD ± 0.80)	9.67 (SD ± 0.74)	0.94	0.052

**Table 2 jcm-12-06852-t002:** Types of microorganisms identified in swabs from the cervical canal and their frequency.

Lp	Pathogen	Group I (N = 637)	Group II (N = 186)
1	*Candida albicans*	302	156
2	*Streptococcus agalactiae*	247	148
3	*Candida glabrata*	42	35
4	*Staphylococcus aureus MSSA*	29	26
5	*Candida tropicalis*	10	14
6	*Enterococcus faecalis*	2	3
7	*Trichomonas vaginalis*	1	0
8	*Klebsiella pneumoniae*	1	1
9	*Candida krusei*	1	1
10	*Escherichia coli*	1	18
11	*Staphylococcus aureus MRSA*	1	2

**Table 3 jcm-12-06852-t003:** Comparison of blood gas, inflammatory markers and bilirubin test results in the analyzed groups of newborns—analysis using the student *t*-test. *p*-values > 0.05—statistically non-significant result.

Variable	Group I (N = 637)	Group II (N = 186)	*p*	Effect Size
pH	7.39 (SD ± 0.02)	7.31 (SD ± 0.07)	0.0001	1.554
BE (mEq/l)	−3.61 (SD ± 2.21)	−6.13 (SD ± 3.72)	0.06	0.824
pO_2_ (mmHg)	46.21 (SD ± 11.01)	45.78 (SD ± 13.11)	0.67	0.036
pCO_2_ (mmHg)	36.75 (SD ± 8.61)	36.66 (SD ± 7.11)	0.88	0.011
HCO_3−_(mmol/L)	23.22 (SD ± 2.64)	19.17 (SD ± 3.23)	0.0001	1.373
Saturation (%)	83.44 (SD ± 6.91)	78.91 (SD ± 13.1)	0.03	0.433
CRP (mg/L)	7.24 (SD ± 6.12)	8.52 (SD ± 8.02)	0.34	0.179
Procalcitonin (ng/mL)	0.67 (SD ± 0.52)	1.74 (SD ± 1.11)	0.01	1.235
Bilirubin (mg/dL)	7.92 (SD ± 3.67)	8.92 (SD ± 4.05)	0.03	0.259
WBC (tys/µL)	12.51 (SD ± 4.35)	19.57 (SD ± 7.48)	0.0001	1.154
Granulocytes (%)	54.67 (SD ± 11.39)	56.42 (SD ± 9.76)	0.15	0.165

**Table 4 jcm-12-06852-t004:** Analysis of disorders of the early adaptation period and procedures performed in a group of newborns from the study group and the control group—analysis with the Chi^2^-Pearson test. *p*-values > 0.05—statistically non-significant result.

Variable	Group I (N = 637)	Group II (N = 186)	*p*	Cramer’s V
Respiratory disorder syndrome	31	85	0.001	0.491
CPAP	21	63	0.001	0.422
Oxygen therapy	8	15	0.02	0.173
Antibiotics	8	13	0.01	0.152
Parenteral nutrition	7	8	0.2	0.100
Jaundice	122	98	0.04	0.317

**Table 5 jcm-12-06852-t005:** Analysis of the type and frequency of infection in newborns in the study groups performed with the Chi^2^–Pearson test. *p*-values > 0.05—statistically non-significant result.

Type of Infection	Group I (N= 8)	Group II (N = 13)	*p*	Cramer’s V
Pneumonia	3	3	0.27	0.155
Urinary tract infection	1	2	0.14	0.040
Infection of undetermined origin	4	8	0.01	0.113

## Data Availability

The data presented in this study are available on request from the corresponding author.

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
