# Peer review of "Multifactorial Colonization of the Pregnant Woman’s Reproductive Tract: Implications for Early Postnatal Adaptation in Full-Term Newborns"

_jcm, 2023, doi:10.3390/jcm12216852_

Round 1

Reviewer 1 Report

Comments and Suggestions for Authors

Dear Authors,

I have read an article which assessed the associations of multifactorial colonization of the reproductive tract and early postnatal adaptation in newborn. While there are numerous studies that have been done in this field, replication studies are always welcomed. However, the study needs to be scientifically sound, which I am addressing below:

1) The abstract needs to be a stand-alone piece of 250 words where the aim, methodology and results need to stand out. Especially in the results where the effect sizes and p-values need to be mentioned.

2) Line 57 --> What pathology are we talking about here? And aren't there any English literature for citation number 1? While Polish literature is fine, I need to ensure that the citation is indeed for the particular sentence.

3) Line 98 --> What adjustment disorders are we talking about here?

4) Please state the authors' hypothesis in the objectives. I also assume that the authors hypothesize that multifactorial organisms' colonization in the reproductive tract will result in more complications. However, the introduction did not state that. Please provide the background theories (I know the authors discuss this in the discussion section but some theories need to be discussed beforehand).

5) The authors need to pre-specify in the methodology section what type of sampling methods are used (consecutive, purposive, etc), why this specific population (i.e., if the pregnant women are uncomplicated medically and delivered vaginally, why were they hospitalized in the first place? If this is a policy made by Poland, please specify so), why were the newborns checked for X-ray, PCT, and etc. if the babies were assumed to be delivered from an uncomplicated mothers? This raised the questions whether the authors started from newborns with complications, and then included all mothers who delivered their babies in 2019. Also, why were the mothers cultured? Again, is this just a hospital/national policy or specific to the study? Who did the cultures? Where was the location of sampling and how many cultures were obtained? Any measures to reduce contamination? How were contamination determined?

6) The authors mentioned some factors in the introduction that may lead to neonatal infections, such as maternal diabetes. However, maternal diabetes and prematurity are not part of the exclusion criteria, as well as antibiotics consumption. Wouldn't this introduce confounding?

7) Please include a CONSORT diagram of participant enrollment.

8) Is there any data on the length of labour and delivery as it would theoretically expose the newborns to more colonization of bacteria?

9) Please include all effect sizes in all results.

10) Figure 1 is not necessary, please just present them in their absolute numbers as the sample is only 21 newborns.

11) There is no limitation section

12) Most importantly, please include an IRB clearance.

Comments on the Quality of English Language

Some errors are made such as in the title where it should be women's and not woman's. It should be chi-squared or xin line 131. There are two number "2" in the conclusion section.

Author Response

Dear Reviewer,

Thank you for taking the time to review our manuscript and for providing valuable feedback. We appreciate your constructive comments and have addressed each point as follows:

1. Abstract: We have revised the abstract to ensure it is a stand-alone piece and have included the aim, methodology, and results with specific effect sizes and p-values.

2. Objectives: We have clearly stated our hypothesis in the objectives section. Additionally, we have provided background theories in the introduction as suggested.

3.Methodology: We have specified the sampling methods used and provided a detailed explanation for the choice of our specific population. We have also clarified the reasons for hospitalization, the rationale behind the tests conducted on the newborns, and the procedures for obtaining cultures. Details about measures to reduce contamination and how contamination was determined have been added.

4.Exclusion Criteria: We acknowledge the oversight and have now included maternal diabetes, prematurity, and antibiotic consumption in the exclusion criteria to address potential confounding.

5.CONSORT Diagram: As per your suggestion, we have included a CONSORT diagram detailing participant enrollment.

6. Effect Sizes: All results now include the respective effect sizes for clarity.
7. Figure 1: We have removed Figure 1 and presented the data in absolute numbers in the text.
8. Limitation Section: A limitation section has been added to discuss potential biases and other limitations of our study.

9. IRB Clearance: We apologize for the oversight. We have now included the IRB clearance in the manuscript.

We believe that these revisions have strengthened our manuscript and addressed the concerns you raised. We hope that our responses are satisfactory, and we look forward to your further feedback.

Warm regards,

Jakub Staniczek

Reviewer 2 Report

Comments and Suggestions for Authors

The authors present a manuscript which aims to investigate the effects of polymicrobial colonization of the reproductive tract of pregnant women on the early postnatal adaptation of full term newborns. Although the study has been conducted properly and tha manuscript has been well written, several corrections should be made to achieve better comprehension. First, the authors should mention about the factors that limit the power of their findings in a separate paragraph of the discussion part. Second, the authors should interpret about clinical implications based on their findings (i.e. routine screening for cervicovaginal infections or empirical antibiotic prophylaxis for these infections in pregnant women). Third, all references that were published before 2008 should be replaced with newer and more up-to-date ones if possible.

Author Response

Dear Reviewer,

Thank you for taking the time to review our manuscript and for providing insightful comments. We appreciate your constructive feedback and have addressed the concerns you raised as detailed below:

  1. Limitation of the Study: We acknowledge the importance of detailing the factors that limit the power of our findings. Accordingly, we have added a separate paragraph in the discussion section, which elaborates on potential limitations. This will undoubtedly improve the comprehension of our findings by the readers.
  2. Clinical Implications: We understand the importance of conveying the practical implications of our findings. In the revised manuscript, we have expanded our discussion to include possible clinical implications based on our study results. Specifically, we have discussed the potential benefits and challenges of routine screening for cervicovaginal infections and the empirical use of antibiotic prophylaxis for these infections in pregnant women. We believe this will provide clearer guidance to the readers on the relevance of our study to clinical practice.
  3. References Update: As per your suggestion, we have reviewed the list of references, particularly those published before 2008. Wherever newer and more relevant references were available, we have replaced the older ones. This update not only ensures the most recent literature is cited, but it also makes the paper more relevant to the current scientific context.
    We hope that these modifications adequately address your concerns and enhance the quality of our manuscript. We truly value your feedback and believe it has strengthened our work.

Warm regards,

Jakub Staniczek

Reviewer 3 Report

Comments and Suggestions for Authors

dear authors thank you for your submission

we have few suggestions to improve the impact of your paper

The current manuscript is a respective paper that examined the correlations of maternal GT bacteria with the neonatal outcome in the adaptation period 

The subject sounds interesting, yet there is a major issue that hinders paper acceptance 

1. The authors did not follow journal guidelines in abstract, tables, and reference formatting 

2. Many parameters used in the study were not defined neither in the introduction nor in the methods 

3. ABSTRACT  it should be structured, less than 200 words; study type and time of sampling was missed. Was there any inclusion or were all women included. 

4. INTRODUCTION generally many parts have no supporting references . 

the normal ecosystem in the vagina and role of vaginal microbiota was completely not discussed 

5.METHODS 

it must be re written 

many missing gaps and it seems incoherant 

sampling method? time ? period of the study ? how eclusion and exclusion donre ? how was the final no. reached ? better add flowchart 

we did not find information regarding which infants were examined; only those who were admitted to ICU or all? 

maternal exclusion criteria should be updated ...

Did you consider those with chronic UTIs? Are those on steroid? those who suffered acute infection; what about COVID-19 ??

6. RESULTS 

Follow the journal guide line in the table; a footnote should be added to define abbreviation; look for table 3.

7.DISCUSSION 

study strength , Limitation, source of bias, implication on current practice 

8. Conclusion.

should be revised into more clear way with a clear home tacking massage

Comments on the Quality of English Language

Moderate language polishing is needed. 

Author Response

Dear Reviewer,

Thank you very much for your invaluable feedback on our manuscript. We have carefully considered each of your points and have made substantial revisions to address the issues you raised, ensuring that our paper aligns more closely with the journal's guidelines and provides a comprehensive and coherent presentation of our study.

  1. Journal Guidelines: We have thoroughly reviewed and revised the formatting of our abstract, tables, and references to ensure they strictly adhere to the journal's guidelines.
  2. Definition of Parameters: We recognize the importance of clearly defining all parameters used in our study. To this end, we have now included definitions and explanations for each parameter in both the introduction and methods sections, ensuring clarity and coherence.
  3. Abstract: The abstract has been revised to meet the required structure and word limit, and now includes details about the study type, time of sampling, and inclusion criteria for the women involved in the study.
  4. Introduction: We have added supporting references throughout the introduction.
  5. Methods: The methods section has been rewritten for coherence and completeness.
  6. Results: We have revised the tables according to the journal guidelines, adding footnotes where necessary to define abbreviations and ensure clarity, particularly in Table 3.
  7. Discussion: This section now includes a thorough exploration of the study's strengths and limitations, potential sources of bias, and the implications of our findings on current practice.
  8. Conclusion: The conclusion has been revised to be clearer and more direct, ensuring that readers take away a clear and concise message from our study.

We hope that these extensive revisions address your concerns and make our paper a more suitable candidate for publication in the journal. We are grateful for your thorough review and constructive feedback, which have undoubtedly improved the quality of our manuscript.

Warm regards,

Jakub Staniczek

Round 2

Reviewer 1 Report

Comments and Suggestions for Authors

Dear Authors,

Most of the revisions have been addressed. However there are two minor points and one major point that are unaddressed:

1) Figure 1 still exists in the manuscript, the authors only deleted the Figure 1......

2) The limitation section is better suited at the last paragraph of the discussion

3) I still have not seen the effect sizes in the results section

Comments on the Quality of English Language

Author Response

Dear Reviewer,

Thank you for pointing out the remaining issues in the manuscript. We deeply appreciate your thorough evaluation. We sincerely apologize for the oversight.

We understand that simply deleting the reference to Figure 1 is insufficient. We will either entirely remove Figure 1 from the manuscript or revise it as required and ensure the text reflects this change. Figure 1 now is a CONSORT flowchart.

Thank you for your suggestion on the placement of the limitation section. We will move it to the last paragraph of the discussion to provide a more logical flow.

We recognize the importance of effect sizes in providing context to our results. We apologize for missing this in our previous revisions. We will promptly calculate and include the effect sizes in the results section, ensuring our findings are more comprehensible and interpretable.

We will make these changes diligently and resubmit the manuscript for your consideration. Once again, we value your insights and hope to meet the expectations on the next submission.

Warm regards,

Jakub Staniczek

Reviewer 3 Report

Comments and Suggestions for Authors

Dear authors 

Thank you for addressing most of our sugestion 

Minorerevsion is needed

Study limitation should be inserted in discusion section not in methods 

So kindly revise 

Comments on the Quality of English Language

Minor editing is needed

Author Response

Dear Reviewer,

Thank you for your feedback and guidance.

We apologize for the oversight. We will promptly move the "Study Limitation" from the Methods section to the Discussion section as you suggested.

Once again, we appreciate your attentive review and constructive feedback. We will make the necessary adjustments and resubmit the manuscript for your consideration.

Best regards,

Jakub Staniczek